# Ethanolic Cashew Leaf Extract: Antimicrobial Activity, Mode of Action, and Retardation of Spoilage Bacteria in Refrigerated Nile Tilapia Slices

**DOI:** 10.3390/foods11213461

**Published:** 2022-11-01

**Authors:** Pitima Sinlapapanya, Punnanee Sumpavapol, Nilesh Nirmal, Bin Zhang, Hui Hong, Soottawat Benjakul

**Affiliations:** 1International Center of Excellence in Seafood Science and Innovation, Faculty of Agro-Industry, Prince of Songkla University, Hat Yai, Songkhla 90110, Thailand; 2Program of Food Science and Technology, Faculty of Agro-Industry, Prince of Songkla University, Hat Yai, Songkhla 90110, Thailand; 3Institute of Nutrition, Mahidol University, 999 Phutthamonthon 4 Road, Salaya, Phutthamonthon, Nakhon Pathom 73170, Thailand; 4Key Laboratory of Health Risk Factors for Seafood of Zhejiang Province, College of Food Science and Pharmacy, Zhejiang Ocean University, Zhoushan 316022, China; 5Beijing Laboratory for Food Quality and Safety, College of Food Science and Nutritional Engineering, China Agricultural University, Beijing 100083, China

**Keywords:** cashew leaf extract, RSM, antimicrobial activity, active compounds, cell damage, spoilage bacteria

## Abstract

Phenolic compounds from cashew (*Anacardium occidentale* L.) leaves were extracted using ethanol with the aid of ultrasonication. Three independent variables, including ultrasound amplitude, time, and ethanol concentration, were used for response surface methodology (RSM) along with the central composite design (CCD). Under the optimized condition (70% amplitude; 40 min; 80% ethanol), the extraction yield and total phenolic contents were 24.50% and 431.16 mg GAE/g dry extract, respectively. Cashew leaf extract (CLE) had the lower minimum inhibitory concentration (MIC) and minimum bactericidal concentration (MBC) against *Shewanella* sp. than *P. aeruginosa*. The release of K^+^ and Mg^2+^ ions from damaged cell membranes with a coincidental decrease of TTC dehydrogenase activity were augmented when treated with CLE. In addition, scanning electron microscopic (SEM) image revealed deformations and perforation of cell walls of bacteria treated with CLE. The dominant compounds in CLE were amentoflavone, quercetin, and its glycosides. Based on microbial challenge test, the growth of *P. aeruginosa* and *Shewanella* sp. inoculated in tilapia slices were inhibited by CLE at 400 and 600 ppm within 15 days of refrigerated storage.

## 1. Introduction

Cashew (*Anacardium occidentale* L.) is an economic plant of Thailand, especially in the south. It has been promoted as an industrial plant cultivated in Thailand and Southeast Asia [1]. Generally, the cashew nut has been known as the major edible plant with a high price. Nevertheless, cashew leaf also contains several phytochemicals, especially bioactive phenolic compounds associated with antioxidant and antimicrobial activities. Numerous phytochemicals were present in cashew leaf extract, including phenolic compounds with diverse bioactivities, including antioxidant and antibacterial activities [2]. Ethanolic cashew leaf extract showed antimicrobial activity towards various microorganisms. Active compounds undergo interaction with proteins at the cell wall to form a complex, thus causing the leakage or disruption of the cell wall [3]. The phenolic compounds in cashew leaf have been shown to inhibit the growth of a number of pathogens [4].

Plant extracts from leaves, including chamuang [5], noni [6], and betel leaf [7], have been documented to extend the shelf life of fish and fish products, mainly due to the antimicrobial activity of phenolics present in those extracts [6,7].

The extraction process is one of the crucial steps in isolating the target bioactive compounds, and one of the easiest and most effective extraction methods is ultrasound-assisted extraction [8,9]. Ultrasonication has the ability to create cavitation, which hastens the breakdown of plant cell walls and accelerates the migration of active compounds into the extracting solvent [10]. In addition, numerous parameters influencing the extraction efficacy of phenolics include sonication time extraction, temperature, sample size, solvent, and frequency of the ultrasonic wave [11]. To optimize extraction efficacy, several techniques, especially response surface methodology (RSM), have been widely used [12,13,14]. Cashew leaf extract has been used as an antioxidant to prevent the oxidation of lipids in mayonnaise [15]. Nevertheless, the extract has not been employed as an antimicrobial agent, especially to prolong or maintain the quality of fish and fish products. The aims of this study were (1) to optimize the extraction of phenolic compounds from cashew leaves using RSM, (2) to determine antimicrobial activity, and (3) to study the ability of the extract to prolong the shelf life of Nile tilapia slices via challenge test.

## 2. Materials and Methods

### 2.1. Chemicals

Chemicals were of analytical grade. Ethanol (98%) was procured from Merck (Darmstadt, Germany). Triphenyl-2H tetrazolium chloride (98%) was purchased from AppliChem (Darmstadt, Germany). Gallic acid (≥98%) was acquired from Sigma-Aldrich, Inc. (Chemie GmbH., Steinheim, Germany). All the microbial media were obtained from Himedia (Mumbai, India).

### 2.2. Collection and Preparation of Cashew Leaves

During April and May 2022, leaves of cashew (*Anacardium occidentale*, L.) were collected from trees (15–20 years old) grown in Songkhla province, Thailand. Cashew leaves from the III (apical) to the V (basal) of the twig were used [16]. After cleaning with tap water, the leaves were dried in an oven at 50 °C for 12 h until the moisture content was below 10%. Dried leaves were blended and sieved (80 mesh) [7]. The powder obtained was referred to as “cashew leaf powder (CLP)”, placed in a zip lock bag and kept in a desiccator until extraction.

### 2.3. Preparation of Cashew Leaf Extract (CLE)

#### 2.3.1. Cashew Leaf Extraction

Ultrasonic equipment (Sonics, Model VC750, Sonica & Materials, Inc., Newtown, CT, USA) was used for extraction [15]. A 250 mL beaker was filled with 10 g of leaf powder and 200 mL of ethanol. To keep the temperature of the mixture below 35 °C, an ice bath was used. Thereafter, the mixture was centrifuged at 5000× *g* for 30 min at 4 °C. The supernatants were filtered with filter paper, and the filtrate was evaporated in an Eyela rotary evaporator at 40 °C (Tokyo Rikakikai, Co. Ltd., Tokyo, Japan). To get rid of the remaining ethanol, nitrogen purging was employed [7]. The obtained extract was freeze-dried, placed in an amble bottle and stored at −20 °C [17].

#### 2.3.2. Dechlorophyllization of Cashew Leaf Extract Using Sedimentation Method

To remove chlorophylls from the extract, the sedimentation process was applied [7]. A mixture of distilled water and the extract (1:1, *v*/*v*) was prepared. Sedimentation was conducted at 4 °C for 24 h and the supernatant was then collected via centrifugation at 10,000× *g* (4 °C for 30 min) before being lyophilized. The dried extract was placed in an amble bottle and stored at −20 °C before being analyzed. The powder was named “CLE”.

### 2.4. Analyses

#### 2.4.1. Extraction Yield

The extraction yield was calculated from the weight of dried CLE relative to that of dry leaf powder. It was reported as a percentage.

#### 2.4.2. Total Phenolic Content (TPC)

TPC of CLE was determined using Folin–Ciocalteu’s reagent (FCR). TPC was computed and reported as mg gallic acid equivalent (GAE)/g dry extract [18].

#### 2.4.3. Optimization of Extraction

The extraction of phenolic compounds from cashew leaf powder was done using ethanol at different concentrations with the aid of ultrasound for various extraction times. The RSM and central composite design (CCD) were applied. Twenty runs with six replicates at the center point were adopted for experimental design. Amplitude (*X*_1_, %), time (*X*_2_, min), and ethanol (*X*_3_, %) were the independent variables, and all variables had zero code at their central values. The levels of variables used are presented in both actual and coded versions, as appeared in Table 1. The average of the triplicates served as the response values (*Y*) for each trial. RSM was implemented to the experimental data using version 7.0 of the Design-Expert Statistical package. The optimal condition was found by superimposing the maximum areas of all the responses obtained from all combinations of independent variables.
(1)$$Y=\beta_{0}+\overset{3}{\underset{i=1}{\sum }}\beta_{i}X_{i}+\overset{3}{\underset{i=1}{\sum }}\beta_{ii}X_{i}{}^{2\;}+\overset{3}{\underset{i=1}{\sum }}\overset{3}{\underset{j=2}{\sum }}\beta_{ij}X_{i}X_{j}$$

*Y* are the dependent variables (yield and TPC). *X_i_* and *X_j_* are the coded versions of the independent variables. *β*_0_, *β_i_*, *β_ii_*, and *β_ij_* are the constant, linear coefficient, quadratic coefficient, and interaction coefficient of the model, respectively.

#### 2.4.4. Verification of the Optimum Extraction Condition

The optimal extraction condition was validated for each response. The experimental errors of models were identified as follows:Error (%) = [(Observed value − Predicted value)/Observed value] × 100

#### 2.4.5. Antibacterial Activity of the Extract

##### Bacterial Strains

Gram-negative bacteria causing spoilage in seafood were tested with CLE. The microorganisms used in the study were *Pseudomonas aeruginosa* PSU.SCB.16S.12, which was obtained from Food Safety Laboratory, Prince of Songkhla University, Thailand, and *Shewanella* sp. TBRC 5775 from Thailand National Center for Genetic Engineering and Biotechnology.

##### Minimum Inhibitory Concentration (MIC) and Minimum Bactericidal Concentration (MBC)

Mueller–Hinton broth microdilution was used to assess the MIC and MBC of CLE in a 96-well microplate. The extract (100 µL) was poured into a sterile 96-well plate at various concentrations (0.59–300 mg/mL). Subsequently, 100 µL of bacterial suspension cultured in TSB medium for 16–18 h at a concentration of 1.5 × 10^6^ CFU/mL was added. A bacterial suspension without extracts was used as a positive control, while a negative control was made with only extracts and no bacterial suspension [19]. The 96-well microplate was incubated at 37 °C for 24 to 48 h. To validate the survival of bacteria, 10 µL of suspension was collected from wells, dropped and spread on TSA agar, and the plates were incubated at 37 °C for 24 h. The MIC was the lowest concentration that completely inhibited bacterial growth, whereas MBC was the lowest concentration that had a bactericidal effect (no growth) [20].

##### Time-Kill Kinetics of Extracts

CLE, with the lowest MIC and MBC, were chosen for the bacterial killing efficiency study. The experiment was carried out according to Tagrida and Benjakul [7]. One mL of the 0.5 McFarland bacterial suspension from each growing culture (1.5 × 10^6^ CFU/mL) prepared in TSB was added to CLE solutions (1 mL) at MIC values of 0.5, 1, 2, and 4. The mixtures were incubated at 37 °C for 24 h. The number of surviving cells was counted using the plate count method at 0, 2, 4, 6, 8, 12, and 24 h. The same procedures were used for the control culture (without CLE).

##### Triphenyl-2H Tetrazolium Chloride (TTC) Dehydrogenase Activity

TTC dehydrogenase is a known product of living bacterial cells and can be utilized as a marker of cell viability. The Tagrida and Benjakul [21] method was adopted to measure the TTC dehydrogenase activity of bacterial cells treated with CLE at varying concentrations (0.5, 1, 2 and 4 MIC) and untreated (control) bacteria. The relative activity of TTC dehydrogenase (RA-TTCD) was determined using the equation shown below.
RA-TTCD (%) = AT/AU × 100
where AT and AU represent, respectively, the absorbance at 510 nm of treated and untreated bacterial cells.

##### Potassium (K^+^) and Magnesium (Mg^2+^) Ion Leakage

K^+^ and Mg^2+^ contents liberated from bacterial cells after treatment with CLE at 0.5, 1, 2, and 4 MIC were determined as tailored by Jayeoye, et al. [22]. K^+^ and Mg^2+^ concentrations were analyzed by inductively coupled plasma optical emission spectroscopy (ICP-OES). The content was reported as mg/L.

##### Scanning Electron Microscopy (SEM)

Cell morphology of untreated and CLE-treated bacteria was visualized using a scanning electron microscope (FEI Quanta 400-ESEM FEG, Hillsboro, OR, USA). CLE at 1 MIC was used for cell treatment, while cells without treatment were used as the control.

#### 2.4.6. Analysis of Phenolic Compounds Using Liquid Chromatography/Diode Array Detector/Mass Spectrometer Detector (LC/DAD/MSD)

Liquid chromatography/mass spectrometric (LC/MS) profiling and identification of CLE extracted under the optimized condition was performed following the procedure of Chotphruethipong, Benjakul and Kijroongrojana [16]. The sample was initially separated using an Agilent 1100 series (Agilent Technologies, Waldbronn, Germany) on the LiChroCART Purospher STAR RP-18e column (Merck, Branchburg, NJ, USA) (150 × 4.6 mm, i.d., 5 m). Mobile phases A and B were acetonitrile and 10 mM ammonium formate buffer pH 4 adjusted with formic acid, respectively. The flow rate was 1.0 mL/min, and the temperature was 40 °C. The gradient program was as follows: 100% B constant (0–5 min), 0–20% A (5–10 min), 20% A constant (10–20 min), and 20–40% A (20–60 min). The detection was done at wavelengths of 270, 330, 350, and 370 nm. The positive and negative ionization modes of MS detection were used, and an electrospray ionizing source with nitrogen gas as drying gas was employed. The following parameters were used: 4000 V capillary voltage, 320 °C gas temperature, 13 L/min flow of the drying gas, and 60 psi nebulizer pressure. MSD was used in the SIM (Selected Ion Monitoring) mode for quantitative analysis.

#### 2.4.7. Impact of CLE on the Growth of Microorganisms Inoculated in Nile Tilapia Slices

##### Preparation of Tilapia Slices

Nile tilapia (*Oreochromis niloticus*) bought from a market at Hat Yai, Songkhla, Thailand (around 5 kg; 1.0 ± 0.2 kg per fish) were brought to the laboratory in ice (1:2, *w*/*w*) within 20 min. Clean water was used to wash the fish. Fish were eviscerated, filleted, and deskinned. Fillets were cut into slices having an average thickness of 1.5 ± 0.1 cm (4 × 2 cm^2^).

##### Effect of CLE on Microbial Growth in Tilapia Slices Inoculated with Selected Bacteria

*P. aeruginosa* and *Shewanella* sp. bacterial suspensions (10^8^ CFU/mL) were incubated for 16–18 h in TSB at 37 °C. The bacterial suspensions were diluted to 10^7^ CFU/mL. A final concentration of approximately 10^6^ CFU/g was achieved by dispersing 10 g of the slices with 1 mL of each bacterial suspension [23]. Fish samples were mixed thoroughly under sterile conditions. Subsequently, CLE was added at concentrations of 200, 400, and 600 ppm. As a positive control, samples were treated only with bacterial suspension. Both the treated samples and the control (10 g each) were put in polyethylene trays and covered with shrink film before being kept at 4 °C for 15 days.

All the samples were collected for bacterial enumeration every 3 days for up to 15 days. The samples (10 g) were added with 90 mL of 0.85% saline solution and homogenized at 230 rpm using a stomacher (Seward 400, Bohemia, NY, USA) for 30 s. *P. aeruginosa* and *Shewanella* sp., and counts were determined using Pseudomonas agar base (PAB) and thiosulfate citrate bile salts sucrose (TCBS) agar [24], respectively. The following equation was used to determine log reduction:Log reduction = Log_10_(A) − Log_10_(B)
where A represents the bacterial count on day 0 of storage and B represents the bacterial count at various storage times.

### 2.5. Statistical Analyses

For the whole study, a completely randomized design (CRD) was used. The experiments and analyses were conducted in triplicate. Analysis of variance (ANOVA) was used, and Duncan’s multiple range test (DMRT) was performed for mean comparison by SPSS package (SPSS 22.0 for Windows, SPSS Inc., Chicago, IL, USA) software.

## 3. Results and Discussion

### 3.1. Yield, TPC and Antimicrobial Activities of CLE Prepared Using Ultrasound-Assisted Extraction (UAE) Process

Based on the central composite design (CCD), ethanol concentration, ultrasound amplitude, and time affected the yield and TPC of resulting extracts (Table 2). Extraction yield (from 18.14 to 24.50%) and TPC (from 469.62 to 321.80 mg GAE/mg dry extract) were affected or determined by all parameters used. The highest yield and TPC of 24.50% and 431.16 mg GAE/mg dry extract, respectively, were obtained when 80% ethanol, sonication amplitude of 70% and extraction time of 40 min were employed. The efficacy of UAE for extracting bioactive compounds from plant materials is markedly influenced by the sonication amplitude [13]. The ultrasonic probe produces ultrasonic waves with high intensity when contacting with CLP [25]. CLP distributed in the solvent was affected by the energy generated by the explosion of the microbubble [26]. When cell walls were destroyed, larger pores were created, facilitating the bulk transfer of active compounds to the solvent, thereby improving yield [27].

Direct contact between the ultrasonic probe and CLP increases the physical and mechanical effects known as the cavitation phenomenon, which breaks the cell wall and creates a porous surface, allowing the release of phenolic compounds. This coincided with increases in yield and TPC [14]. Not only the high-intensity ultrasonic waves but also the increasing contact time between the CLP and ethanol resulted in augmented extraction efficacy [28]. The extended extraction period could increase the diffusion of the bioactive compounds from the CLP into the ethanol, hence increasing the yield of phenolic compounds [29]. Ethanol concentration could affect the solubility of bioactive compounds from cashew leaf powder [30]. However, the obtained extracts were subjected to subsequent dechlorophyllization using a sedimentation process in which no solvents were used. This chlorophyll removal might contribute to the lower yield, although the color was improved. Ethanolic extracts from betel leaves and noni leaves were successfully dechlorophyllized with the aid of the sedimentation process, a green process [6,7].

### 3.2. Optimal Condition of UAE for Extraction of CLE Using RSM

Conditions for extraction of CLE as monitored by yield and TPC using RSM were optimized. Independent parameters (amplitude, time, and ethanol) and the dependent parameters (yield and TPC) were set in the range of each parameter and the maximum level in the analytical process, respectively, to achieve the condition rendering the highest yield and TPC. The variables included sonication amplitude (*X*_1_, %), time (*X*_2_, min), and ethanol concentration (*X*_3_, %). Experimental variables based on the CCD experiment at three variation levels (−1, 0, and +1) are shown in Table 1. Extraction yield, TPC, and antimicrobial activity are given in Table 2. The second-order polynomial equation was fitted to the experimental data, in which % extraction yield and TPC had coefficients of analysis (*R*^2^) of 0.8709 and 0.8813, respectively (Table 3). The significant *p*-value for the model was used. ANOVA result showed that at least one of the model parameters could account for the variation in response variables in the experiment. The quality of the fitted model was also validated by a non-significant lack of fit.

#### 3.2.1. Effect of Independent Variables on Extraction Yield

Different parameters showed various impacts on yields. The extraction yield was between 18.14 and 24.50%. The significance of each coefficient was determined using the *p*-value, as shown in Table 3. The amplitude (*X*_1_) and ethanol concentration (*X*_3_) and two quadratic terms (*X*_1_^2^ and *X*_2_^2^) significantly affected the yield of CLE (*p* < 0.05). The contour plots in Figure 1A–C of results showed all the factors improved yield. The extraction yield was increased as variable values increased. With 70% amplitude, 40 min, and 80% ethanol, the maximum yield was gained. The predicted model’s yield value had an *R*^2^ value of 0.8709 (Table 3). Additionally, the model’s *p*-value (0.0021) and non-significant lack of fit (0.2187) indicated good estimations. In general, ultrasound has been applied to increase the efficacy of plant extraction due to the cavitation effect. González-Silva et al. [25] used ultrasound-assisted extraction (UAE) of polyphenols from *Psidium cattleianum* (PC) leaves to improve the yield extraction. Furthermore, Chotphruethipong et al. [16] used ultrasound for the extraction of phenolics from cashew leaf, and the yield was increased. The extract showed antioxidant activity when tested by several assays. The following equation based on all independent variables on the extraction yield of CLE was shown as follows:Yield = 21.48 + 0.94*X*_1_ + 0.24*X*_2_ + 0.40*X*_3_ + 0.17*X*_1_*X*_2_ + 0.44*X*_1_*X*_3_ − 0.020*X*_2_*X*_3_ − 0.41*X*_1_^2^ + 0.36*X*_2_^2^ + 0.29*X*_3_^2^(2)

#### 3.2.2. Effect of Independent Variables on TPC

TPC of CLE under various extraction conditions ranged from 321.80 to 469.62 mg GAE/g dry extract. The significance of each coefficient was determined using *p*-value. The data in Table 3 demonstrated that ethanol concentration (*X*_3_), the interaction between ethanol concentration (*X*_3_) and extraction time (*X*_2_), and two quadratic terms (*X*_2_^2^ and *X*_3_^2^) significantly affected TPC of CLE (*p* < 0.05). Furthermore, the ethanol concentration was the major factor influencing TPC. The relationship of all factors affecting the increase in TPC of CLE is shown in Figure 2A–C. However, longer extraction times more likely accelerated the breakdown of phenolic substances [16]. Ethanol is one of the most widely used solvents because it can penetrate plant tissue easily and escalates the extraction process [31]. Similarly, Chotphruethipong, Benjakul and Kijroongrojana [16] found the highest TPC in cashew leaf extract when 80% ethanol was used as the extracting medium. The following equation showed the effect of independent variables on TPC of CLE:TPC = 369.25 + 5.37*X*_1_ + 9.35*X*_2_ + 29.55*X*_3_ − 8.54*X*_1_*X*_2_ − 10.23*X*_1_*X*_3_ + 14.27*X*_2_*X*_3_ + 3.36*X*_1_^2^ + 13.47*X*_2_^2^ + 11.06*X*_3_^2^(3)

#### 3.2.3. Effect of Independent Variables on Antimicrobial Activity

Antimicrobial activity of different CLE were obtained as expressed by varying MIC and MBC of 2 bacteria treated (Table 2). Those two bacteria namely *P. aeruginosa* and *Shewanella* sp. are generally acknowledged as the major cause of spoilage of fish and other seafood [32,33]. Those specific perishable organisms (SSOs) can grow quickly and produce metabolites at low temperatures [34]. *P. aeruginosa* was found in tilapia sashimi [35], squid rings [36], and smoked salmon [37]. For rainbow trout fillets, *Shewanella* sp. was found to be the most prevalent genera [38]. It was also the main spoiler of Pacific white shrimp [39] and salmon fillets [40].

Different extraction conditions significantly influenced the antimicrobial activity of the CLE obtained (*p* < 0.05). Bakkaloglu, Arici and Karasu [12] documented that the antibacterial activity of propolis extract using ultrasound was governed by the type of solvent, in which ethanolic extract had the highest antimicrobial activity. Pobiega, et al. [41] also found that the increase in ultrasonication time had the profound inhibitory effect on microorganisms. High levels of phenolic compounds can cause inactivation of bacterial enzymes and the death of bacterial cells [12].

### 3.3. Verification of the Optimum Condition

The statistical model and regression equation for the ideal condition (80% ethanol, 40 min, and 70% amplitude) were validated. The following expected values were found under the optimal condition: 23.90% and 436.90 mg GAE/g dry extract for yield and TPC, respectively. The yield and TPC observed values were 21.98 ± 0.74% (error = 8.03%) and 430.24 ± 7.73 mg GAE/g (error = 1.53%), respectively. Similar anticipated and observed values proved the statistical model’s acceptance and applicability for optimizing cashew leaf extraction. As a result, under optimal condition, the successful extraction of phenolic components from cashew leaves was achieved. Chotphruethipong, Benjakul and Kijroongrojna [16] also validated the optimized condition for extraction of antioxidative phenolic compounds from cashew leaves using response surface methodology as evidenced by non-significant lack of fit. Also, the similar predicted and observed values were also attained.

### 3.4. Time-Kill Kinetics Analysis

A time-kill kinetics to reveal the antibacterial effectiveness of CLE against *P. aeruginosa* and *Shewanella* sp. at various doses are depicted in Figure 3. A decreased growth rate was found when CLE at 0.5 MIC was used, compared to that of control after 24 h of incubation (*p* < 0.05). CLE at low concentrations exhibited fairly poor antibacterial action, as evidenced by continuous growth of both bacteria tested. When CLE concentrations increased, especially above 0.5 MIC, the decrease in counts for both bacteria was noticeable in a dose-dependent manner. No *P. aeruginosa* was detected after 24 h when treated with CLE at 4 MIC. Complete inactivation of *Shewanella* sp. was obtained when treated with CLE at levels of 4 MIC, 2 MIC and 1 MIC after 4, 12 and 12 h of exposure. When the bacterial count drops by 3 logs of the initial inoculum concentration, a substance is considered “bactericidal” [42]. After 24 h of incubation, CLE at 2.34 mg/mL (MIC) demonstrated bacteriostatic activity toward *P. aeruginosa*, in which a decrease of 2.44 CFU/mL was achieved. For CLE at 0.59 mg/mL, a reduction of 8.72 log CFU/mL could fully kill *Shewanella* sp. The findings revealed that the extract was bactericidal to the test organisms at 1 MIC, 2 MIC, and 4 MIC. However, 0.5 MIC showed a bacteriostatic effect. Betel leaf ethanolic extract at 2 MIC showed antimicrobial activity, depending on the time all bacteria tested, including *S. aureus*, *E. coli*, *P. aeruginosa* and *S. sonnei* exposed to the extract. With a longer exposure time (0–24 h), a higher reduction of bacteria was obtained [7]. Overall, both the concentrations and the exposure time had a profound impact on the growth of bacteria treated by CLE.

### 3.5. Triphenyl-2H Tetrazolium Chloride (TTC) Dehydrogenase Activity

TTC (2,3,5-Triphenyl Tetrazolium Chloride) is a small MW substance, which can be taken up by living bacteria via their cell walls or membranes, where it is converted into the insoluble red product, namely 2,3,5-triphenylformazan (TF) [43]. When comparing bacterial survival, TF generated can be utilized to indicate dehydrogenase activity [42]. The decrease of TTC dehydrogenase activity indicated that treatment with CLE had an impact on the inactivation of bacteria (Figure 4). For *P. aeruginosa* and *Shewanella* sp., TTC dehydrogenase activities of CLE decreased with increasing MIC. At 4 MIC, TTC dehydrogenase was decreased by 88.37% and 85.36% for *P. aeruginosa* and *Shewanella* sp., respectively. The relative activity of untreated bacterial cells was from 90.7 to 92.68%. The treatment of *P. aeruginosa* and *Shewanella* sp. with CLE at increasing concentrations reduced TTC-dehydrogenase. This suggested the inhibition of anabolism and respiration, thus eventually causing the death of bacteria [21]. In addition, liposomes loaded with betel leaf extract could decrease TTC dehydrogenase of *S. aureus*, *E. coli*, *P. aeruginosa* and *S. sonnei*, indicating its antimicrobial activity [21]. Therefore, the drop in TTC dehydrogenase activity reconfirmed the bactericidal action of CLE toward both bacteria tested.

### 3.6. Potassium (K^+^) and Magnesium (Mg^2+^) Ion Leakage

Leakage of potassium (K^+^) and magnesium (Mg^2+^) ions from the damaged membrane of *P. aeruginosa* and *Shewanella* sp. is presented in Table 4. CLE treatment resulted in the release of K^+^ and Mg^2+^ from bacterial cells in comparison to the control (untreated). Treatments with CLE at higher concentrations caused the augmented leakage of K^+^ and Mg^2+^ (*p* < 0.05). Potassium is an important cation in cells, and its assimilation is essential for all living organisms for intracellular enzymatic activity. It acts as the second messenger inside the cell and involves maintaining a constant internal pH and membrane potential [44]. Magnesium ions play a role in peptidoglycan synthesis to strengthen the cell wall and the prevention of cell lysis related to cell viability [21,45]. The permeability of various molecules such as water, nutrients, waste, and ions, is a function of a cytoplasmic membrane that encapsulates bacterial cells [46]. Several antimicrobial compounds have been found to interact with the cell membrane of the bacterial cytoplasm, causing pores [47]. This results in a loss of integrity and release of the contents from bacterial cells, leading to bacterial death. Liposomes loaded with betel leaf extract at 2 MIC could induce the leakage of both K^+^ and Mg^2+^ of *S. aureus, E. coli, P. aeruginosa* and *S. sonnei*. This confirmed the bacterial membrane disruption associated with cell death [21].

### 3.7. Effect of CLE on Morphology of Bacterial Cells

The effect of CLE at 1 MIC on the selected bacterial morphology using scanning electron microscopy (×20,000) was illustrated in Figure 5. *P. aeruginosa* and *Shewanella* sp. (untreated) had smooth surfaces and were rod-shaped, as shown in Figure 5. The treatment of *P. aeruginosa* and *Shewanella* sp. using CLE resulted in changes in cell shape, in which cells became perforated, distorted, and smaller, as indicated by the arrow signs. Olatunde, Benjakul and Vongkamjan [42] and Tagrida and Benjakul [7] also found that the extracts of coconut husk and betel leaf extracts caused the injury of *P. aerugenosa*, *V. paraheamyticus*, *E. coli*, *L. monocytogenes*, and *S. aureus* cells. This alteration had a profound impact on their capacity for cell division, repair, and metabolism, thus bringing about lowered antibacterial resistance and eventual death.

### 3.8. LC/DAD/MSD Analysis of Phenolic Compounds

Phenolic compounds of cashew leaf extract under the optimized extracting condition are shown in Table 5. Analysis was performed in negative mode using LC/PAD/MSD. The negative mode has generally higher sensitivity compared to the positive mode. Due to the ability to rapidly produce negative ions, LC/MS operating in the negative mode was more sensitive and can be linked to their propensity to produce negative ions quickly for profiling the phenolic and polyphenolic group of components [32,48]. Phenolic compounds were identified based on the possible molecular formula, *m*/*z*, and abundance. Amentoflavone was the most prominent compound, followed by quercetin 3-galactoside, quercetin 3-(2-galloylglucoside), tricetin 3′-xyloside, and quercetin 3-(2″-galloyl-alpha-L-arabinopyranoside), respectively. Amentoflavone has been known to possess antimicrobial activity, as reported by Yu, et al. [49]. The other compounds also showed antimicrobial activity [50].

The phenolic compounds found in this extract were similar to those reported by Sae-Leaw and Benjakul [2] and Chotphruethipong and Benjakul [15] for the cashew leaf extract. It was found that some phenolics linked with sugar in the form of glycoside. Free phenolic acids are rare since they are bonded as glycosides or esters [51]. However, differences in phenolic compounds are governed by several factors, including harvesting time, genotype, maturity, climate, soil, extraction methods, as well as analysis methods [52].

For positive mode, quercetin was dominant, followed by kiwiionoside (Table 6). Quercetin was also found as glycosides, namely quercetin 3-(2-galloylglucoside), quercetin 3-arabinoside, quercetin 3-(2″-galloyl-alpha-L-arabinopyranoside), and quercetin 3-(2″-galloylrhamnoside). Quercetin was reported to exhibit antimicrobial activity [53]. Thus, the cashew leaf extract was shown to be a promising source of antimicrobial phenolic compounds. LC/MS results confirmed the role of different phenolic compounds in inactivating several bacteria (Table 5 and Table 6).

### 3.9. Effect of CLE on Reduction of Spoilage Microorganisms in Nile Tilapia Slices Using Microbial Challenge Test

The effectiveness of CLE against *P. aeruginosa* and *Shewanella* sp. inoculated in tilapia slices at various concentrations (200, 400, and 600 ppm) is shown in Figure 6. The bacterial count on the first day of storage was 5.74 and 6.15 log CFU/g for *P. aeruginosa* and *Shewanella* sp., respectively, as shown in Figure 6. A large inoculum was used at the beginning of the test to guarantee the proliferation of the target bacteria and to be easy to monitor the reduction [54]. On day 3, the number of controls for each challenged bacteria increased, while samples treated with CLE at various concentrations showed a reduction of the challenged bacteria. CLE at 200 ppm showed lower effectiveness in the retardation of bacterial growth, whereas CLE at 400 and 600 ppm provided the maximum inhibition effect toward the challenged bacteria (*p* < 0.05). This might be related to the antibacterial effects of CLE, which were mediated by polyphenolics, mainly amentoflavone and quercetin 3-galactoside [49,55]. By preventing bacterial growth during low-temperature storage, shelf life could be extended [56].

The difference in the log CFU/g of the evaluated microorganisms between the sample without and with CLE treatment is referred to as “log reduction,” and it provides information about how well an antimicrobial agent works to reduce a load of microorganisms [21]. The sensitivity of the target bacteria also plays an important role in bacterial inhibition. In general, *P. aeroginosa* was more resistant than *Shewanella* sp. due to its ability to form biofilms that act as a protective barrier toward the permeability of antimicrobial agents [57]. On day 9, the highest log reduction (*p* < 0.05) was observed for the sample treated with CLE at 400 and 600 ppm, which the ranged of 1.67–1.85 and 2.50–2.75 log reduction for *P. aeruginosa* and *Shewanella* sp., respectively. This demonstrated the satisfactory efficacy in controlling the proliferation of the bacteria tested and reflecting the antibacterial action of CLE. The CLE treatment became less efficient, as ascertained by lowered log reduction on day 12. The decomposition of active components in CLE might result in a reduction in its efficacy and lead to an increase in bacterial load. Nile tilapia treated with betel leaf ethanolic extract at 400 and 600 ppm could be stored at 4 °C for up to 9 days, in which total viable count was still under the limit [7]. In addition, the shelf life of striped catfish slices added with noni leaf ethanolic extract at 400 ppm was extended to 9 days when kept under refrigerated conditions [6].

## 4. Conclusions

Extraction conditions for phenolic compounds with high yield, TPC, and antimicrobial activity from cashew leaves were optimized using RSM. The best extraction conditions involved the extraction using ultrasound at 70% amplitude and 80% ethanol concentration for 40 min. CLE-treated bacterial cells had lower TTC dehydrogenase activity, and K^+^ and Mg^2+^ leakage. Destruction of bacterial cell walls resulted in a deformed shape and porosity in the cell wall. CLE also retarded spoilage microbial growth in Nile tilapia slices within 15 days of storage at 4 °C. The extract could therefore be used as a natural antimicrobial agent for extending the shelf life of perishable food, especially fish and shellfish.

## Figures and Tables

**Figure 1 foods-11-03461-f001:**
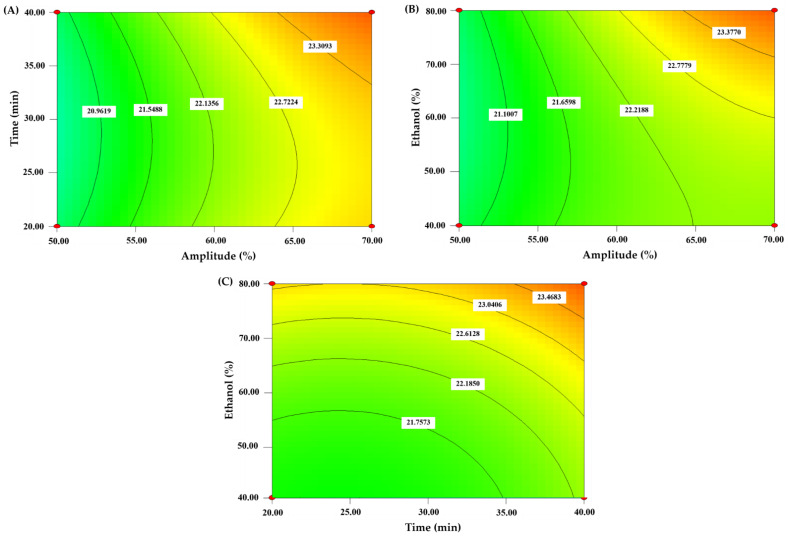
The contour plots of yield from CLE extraction as influenced by the chosen independent factors. (**A**) time and amplitude; (**B**) ethanol and amplitude; (**C**) ethanol and time.

**Figure 2 foods-11-03461-f002:**
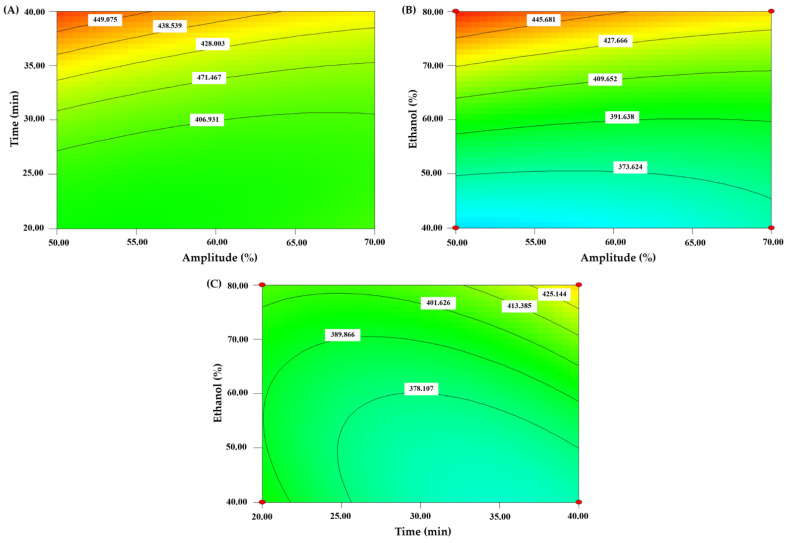
The contour plots of TPC from CLE extraction as influenced by the chosen independent factors. (**A**) time and amplitude; (**B**) ethanol and amplitude; (**C**) ethanol and time.

**Figure 3 foods-11-03461-f003:**
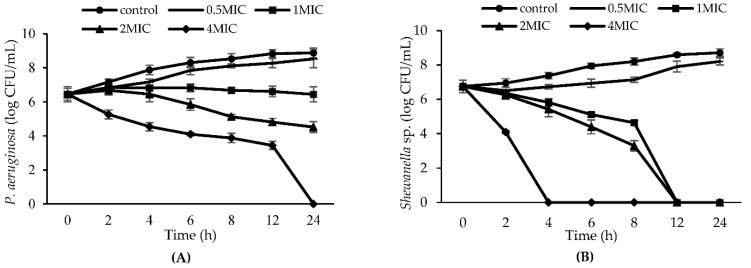
Time-kill curves of *P. aeruginosa* (**A**) and *Shewanella* sp. (**B**) without and with CLE treatment at different concentrations. Bars represent standard deviation (*n* = 3). MIC: Minimum Inhibitory Concentration.

**Figure 4 foods-11-03461-f004:**
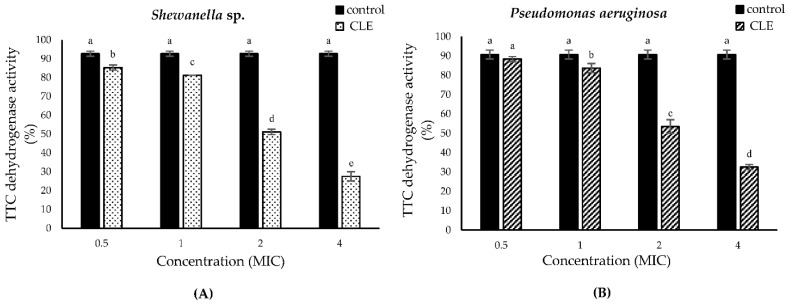
Triphenyl-2H tetrazolium chloride (TTC) dehydrogenase activity of *P. aeruginosa* (**A**) and *Shewanella* sp. (**B**) without (control) and with CLE treatment at different concentrations. Bar represent standard deviation (*n* = 3). Different lowercase letters on the bars denote significant difference (*p* < 0.05).

**Figure 5 foods-11-03461-f005:**
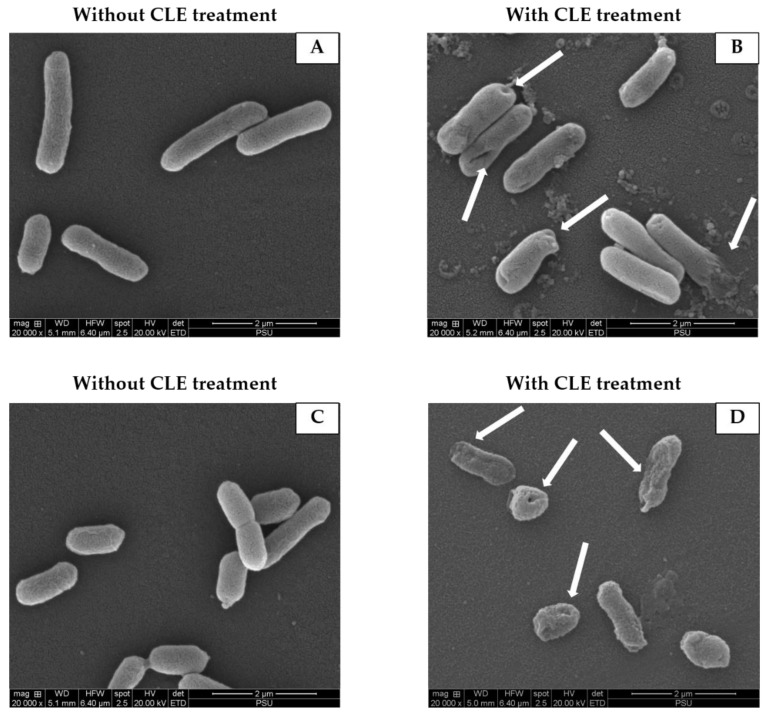
Scanning electron micrographs of *P. aerugenosa* (**A**,**B**) and *Shewanella* sp. (**C**,**D**) without and with CLE treatment at 1 MIC using CLE. Magnification: ×20,000. Arrow signs indicate pore or disrupt membrane.

**Figure 6 foods-11-03461-f006:**
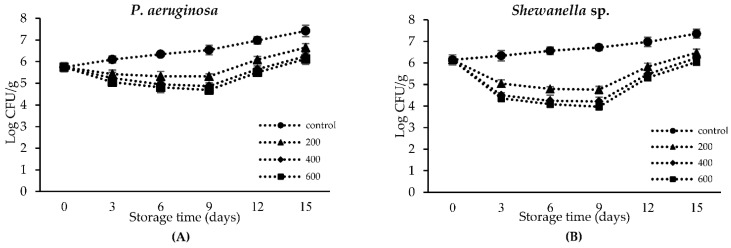
Effectiveness of CLE at different concentrations (200, 400, and 600 ppm) against *P. aeruginosa* (**A**) and *Shewanella* sp. (**B**) counts in tilapia slices during refrigerated storage. Challenge test was used in this study. Bars represent the standard deviation (*n* = 3).

**Table 1 foods-11-03461-t001:** Central composite design for extraction of CLE.

Run Order	Coded Levels			Actual Levels		
	Amplitude (*X*_1_)	Time (*X*_2_)	Ethanol (*X*_3_)	Amplitude (%)	Time (min)	Ethanol (%)
1	1	−1	1	70	20	80
2	0	+1.68	0	60	46.82	60
3	1	−1	−1	70	20	40
4	0	0	−1.68	60	30	26.36
5	−1	−1	−1	50	20	40
6	+1.68	0	0	76.82	30	60
7	0	0	0	60	30	60
8	1	1	−1	70	40	40
9	0	0	0	60	30	60
10	0	0	0	60	30	60
11	0	−1.68	0	60	13.18	60
12	1	1	1	70	40	80
13	0	0	+1.68	60	30	93.64
14	−1	1	−1	50	40	40
15	−1.68	0	0	43.18	30	60
16	−1	1	1	50	40	80
17	0	0	0	60	30	60
18	0	0	0	60	30	60
19	0	0	0	60	30	60
20	−1	−1	1	50	20	80

**Table 2 foods-11-03461-t002:** Yield, total phenolic contents (TPC) and antimicrobial activities of CLE under different extracting conditions using central composite design (CCD).

Run Order	Amplitude (%)	Time (min)	Ethanol (%)	Yield (%)	TPC (mg GAE/g Dry Extract)	PA (mg/mL)		SP (mg/mL)	
						MIC	MBC	MIC	MBC
1	70	20	80	23.04 ± 0.15 ^b^*	397.13 ± 4.42 ^c^	2.34 ^a^	9.38 ^a^	0.59 ^a^	1.17 ^a^
2	60	46.82	60	22.26 ± 0.21 ^c^	414.55 ± 12.86 ^bc^	2.34 ^a^	18.75 ^b^	0.59 ^a^	1.17 ^a^
3	70	20	40	21.62 ± 0.11 ^e^	408.13 ± 4.64 ^c^	2.34 ^a^	18.75 ^b^	1.17 ^b^	9.38 ^c^
4	60	30	26.36	21.44 ± 0.19 ^f^	321.80 ± 12.73 ^g^	2.34 ^a^	9.38 ^a^	1.17 ^b^	9.38 ^c^
5	50	20	40	20.66 ± 0.33 ^gh^	360.92 ± 1.85 ^e^	4.69 ^b^	18.75 ^b^	0.59 ^a^	2.34 ^b^
6	76.82	30	60	21.70 ± 0.14 ^e^	385.11 ± 14.28 ^d^	4.69 ^b^	9.38 ^a^	0.59 ^a^	2.34 ^b^
7	60	30	60	21.39 ± 0.38 ^ef^	375.02 ± 10.34 ^d^	4.69 ^b^	9.38 ^a^	0.59 ^a^	9.38 ^c^
8	70	40	40	22.24 ± 0.29 ^cd^	383.59 ± 13.60 ^d^	4.69 ^b^	9.38 ^a^	0.59 ^a^	2.34 ^b^
9	60	30	60	21.62 ± 0.14 ^e^	378.10 ± 16.68 ^de^	4.69 ^b^	9.38 ^a^	1.17 ^b^	9.38 ^c^
10	60	30	60	20.65 ± 0.26 ^h^	375.10 ± 7.49 ^d^	4.69 ^b^	18.75 ^b^	0.59 ^a^	1.17 ^a^
11	60	13.18	60	21.98 ± 0.32 ^de^	390.54 ± 3.57 ^d^	2.34 ^a^	37.50 ^c^	0.59 ^a^	9.38 ^c^
12	70	40	80	24.50 ± 0.11 ^a^	431.16 ± 15.44 ^b^	2.34 ^a^	9.38 ^a^	0.59 ^a^	1.17 ^a^
13	60	30	93.64	22.43 ± 0.12 ^c^	469.62 ± 1.82 ^a^	2.34 ^a^	9.38 ^a^	0.59 ^a^	2.34 ^b^
14	50	40	40	21.51 ± 0.18 ^e^	372.04 ± 6.91 ^e^	2.34 ^a^	75.00 ^d^	0.59 ^a^	2.34 ^b^
15	43.18	30	60	18.14 ± 0.10 ^i^	362.74 ± 0.87 ^e^	2.34 ^a^	37.50 ^c^	0.59 ^a^	1.17 ^a^
16	50	40	80	21.09 ± 0.13 ^fg^	459.01 ± 12.02 ^a^	2.34 ^a^	37.50 ^c^	1.17 ^b^	1.17 ^a^
17	60	30	60	22.02 ± 0.15 ^d^	362.52 ± 5.34 ^e^	4.69 ^b^	9.38 ^a^	0.59 ^a^	1.17 ^a^
18	60	30	60	21.53 ± 0.27 ^e^	352.97 ± 3.61 ^f^	2.34 ^a^	9.38 ^a^	0.59 ^a^	1.17 ^a^
19	60	30	60	21.78 ± 0.15 ^e^	373.42 ± 1.81 ^e^	2.34 ^a^	9.38 ^a^	0.59 ^a^	1.17 ^a^
20	50	20	80	21.24 ± 0.23 ^f^	392.34 ± 11.47 ^cd^	4.69 ^b^	75.00 ^d^	0.59 ^a^	1.17 ^a^

PA: *Pseudomonas aeruginosa*, and SP: *Shewanella* sp. Different lowercase superscripts within the same column indicate significant differences (*p* < 0.05). * Value is mean ± SD (*n* = 3).

**Table 3 foods-11-03461-t003:** The regression coefficients of the second-order polynomial models of the predicted for yield and total phenolic contents (TPC) of CLE.

Regression Coefficients (*β*)	Yield		TPC	
	% Yield	*p*-Value	TPC	*p*-Value
Intercept				
*β* _0_	21.48		369.25	
Linear				
*β* _1_	0.94	0.0001	5.37	0.2608
*β* _2_	0.24	0.1646	9.35	0.0647
*β* _3_	0.40	0.0294	29.55	<0.0001
Cross product				
*β* _12_	0.17	0.4250	−8.54	0.1775
*β* _13_	0.44	0.0599	−10.23	0.1129
*β* _23_	−0.02	0.9251	14.27	0.0358
Quadratic				
*β* _11_	−0.41	0.0226	3.36	0.4617
*β* _22_	0.36	0.0412	13.47	0.0118
*β* _23_	0.29	0.0841	11.06	0.0303
*R* ^2^	0.8709		0.8813	
Lack of fit	0.2187		0.0539	
*p*-value	0.0021		0.0014	

**Table 4 foods-11-03461-t004:** Content of potassium (K^+^) and magnesium (Mg^2+^) ions released from *P. aeruginosa* and *Shewanella* sp. cells treated with CLE at varying concentrations.

Concentration (MIC)	*P. aeruginosa*		*Shewanella* sp.	
	K^+^ (mg/L)	Mg^2+^ (mg/L)	K^+^ (mg/L)	Mg^2+^ (mg/L)
Control	ND	<0.05 ^e^	5.26 ± 0.39 ^d^	0.12 ± 0.01 ^e^
0.5	25.07 ± 1.03 ^d^*	1.39 ± 0.03 ^d^	5.38 ± 0.36 ^cd^	0.36 ± 0.01 ^d^
1	53.21 ± 1.82 ^c^	2.13 ± 0.04 ^c^	6.34 ± 0.63 ^c^	0.66 ± 0.01 ^c^
2	103.00 ± 1.60 ^b^	4.28 ± 0.09 ^b^	33.27 ± 0.44 ^b^	1.30 ± 0.08 ^b^
4	213.10 ± 2.80 ^a^	8.95 ± 0.02 ^a^	65.11 ± 2.38 ^a^	2.86 ± 0.03 ^a^

Different lowercase superscripts within the same column indicate significant differences (*p* < 0.05). Abbreviation: ND: not detected. * Value is mean ± SD (*n* = 3).

**Table 5 foods-11-03461-t005:** Phenolic compounds in CLE extracted under optimized conditions were analyzed by LC/DAD/MSD in negative mode.

Identified Compounds	Formula	*m*/*z*	Mass	R_t_ (min)	Abundance (×10^6^)
Amentoflavone	C_30_H_18_O_10_	537.08	538.09	32.62	16.35
Quercetin 3-galactoside	C_21_H_20_O_12_	463.09	464.10	17.76	14.48
Quercetin 3-(2 galloylglucoside)	C_28_H_24_O_16_	615.10	616.11	18.18	11.94
Tricetin 3′-xyloside	C_20_H_18_O_11_	433.08	434.08	18.82	11.24
Quercetin 3-(2″-galloyl-alpha-L-arabinopyranoside)	C_27_H_22_O_15_	585.09	586.10	20.68	10.48
Kaempferol 4′-glucoside	C_21_H_20_O_11_	447.09	448.10	20.07	9.14
(±)-Catechin	C_15_H_14_O_6_	289.07	290.08	10.63	8.15
4-Glucogallic acid	C_13_H_16_O_10_	331.07	332.07	3.40	7.95
Quercetin 3-(2″-galloylrhamnoside)	C_28_H_24_O_15_	599.10	600.11	24.20	6.32
Epigallocatechin	C_15_H_14_O_7_	305.07	306.07	6.46	5.91
Citbismine C	C_37_H_36_N_2_O_11_	683.23	684.23	2.02	2.55
(−)-Catechin gallate	C_22_H_18_O_10_	441.08	442.09	17.63	2.08
Quercetin	C_15_H_10_O_7_	301.04	302.04	20.65	1.30
Epigallocatechin Gallate	C_22_H_18_O_11_	457.08	458.08	13.36	1.24
Gallic acid	C_7_H_6_O_5_	169.01	170.02	4.28	1.13
Kiwiionoside	C_19_H_34_O_9_	465.23	406.22	7.88	1.08

**Table 6 foods-11-03461-t006:** Phenolic compounds in CLE extracted under optimized extracting condition analyzed by LC/DAD/MSD in positive mode.

Identified Compounds	Formula	*m*/*z*	Mass	R_t_ (min)	Abundance (×10^6^)
Quercetin	C_15_H_10_O_7_	303.05	302.04	17.74	4.60
Kiwiionoside	C_19_H_34_O_9_	429.21	406.22	7.89	3.94
Quercetin 3-(2-galloylglucoside)	C_28_H_24_O_16_	639.10	616.11	18.14	3.13
Quercetin 3-arabinoside	C_20_H_18_O_11_	457.08	434.09	18.85	2.42
Amentoflavone	C_30_H_18_O_10_	539.10	538.09	32.68	1.43
Quercetin 3-(2″-galloyl-alpha-L-arabinopyranoside)	C_27_H_22_O_15_	609.09	586.10	23.59	1.39
Epicatechin	C_15_H_14_O_6_	291.09	290.08	10.53	1.13
Sambacin	C_26_H_36_O_12_	563.21	540.22	14.75	0.60
Chlorogenic Acid	C_16_H_18_O_9_	355.10	354.10	6.32	0.44
ent-Epicatechin-(4 alpha-> 8)-ent-epicatechin 3-gallate	C_37_H_30_O_16_	731.16	730.15	14.54	0.37
Quercetin 3-(2″-galloylrhamnoside)	C_28_H_24_O_15_	623.10	600.11	20.50	0.33
Luteolin	C_15_H_10_O_6_	287.06	286.05	22.41	0.32
(1S,2R,4R,8S)-p-Menthane-2,8,9-triol 9-glucoside	C_16_H_30_O_8_	373.18	350.20	10.96	0.24
Prosopinine	C_16_H_33_NO_3_	288.25	287.25	37.40	0.17
Epigallocatechin	C_15_H_14_O_7_	307.08	306.08	6.47	0.11

## Data Availability

Data are provided in the article.

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
