# Peer review of "Ethanolic Cashew Leaf Extract: Antimicrobial Activity, Mode of Action, and Retardation of Spoilage Bacteria in Refrigerated Nile Tilapia Slices"

_foods, 2022, doi:10.3390/foods11213461_

Round 1
Reviewer 1 Report
Although this work is presented in a systematic way with explained used methods and obtained results, the discussion part needs to be refined. In chapters 3.1., 3.2.1., 3.2.2., 3.3., 3.4., 3.5., 3.6. and 3.9. there is no comparison with data from the literature. In the file below there are several things listed that have to be revise.
Line 27: abbreviation SEM has to be explained Line 66: Ethanol was procured Merck….please revise this (e.g. Ethanol was produced by Merck..)
Line 114: Y is the dependent variables…as Y stands for few variables, should it be Y are dependent variables? Line 140: please replace this °C with correct one
Line 172: abbreviations LC/DADMSD and LC/MS have to be explained
Lines 174 and 346: numbers of references in brackets and brackets are written in different type of letters
Line 192: cm2 should be replaced with cm2
Line 198: reference should be converted in the number of the reference and according to that, all the other references need to be checked
Line 220: This sentence is unclear, please rephrase it. The begging of this section needs to be complemented, meaning there has to be a little entry in the results
Line 381: make a space between P. aeruginosa
Line 470 and 473: there are no dots at the end of those two titles and in the other titles of figures and tables the authors have put the dot at the end
References: somewhere the full name of journal is written and somewhere only abbreviation
Why there are no dots after the number of the subchapters?

Author Response
Responses to reviewer
Comments and Suggestions for Authors
Although this work is presented in a systematic way with explained used methods and obtained results, the discussion part needs to be refined. In chapters 3.1., 3.2.1., 3.2.2., 3.3., 3.4., 3.5., 3.6. and 3.9. there is no comparison with data from the literature. In the file below there are several things listed that have to be revise.
***Thank you very much for valuable comments and suggestions. The comparison of the data has been made with other literatures for all sections raised by the reviewer. Those are highlighted in yellow.
Line 27: abbreviation SEM has to be explained
***Definition of SEM has been provided. Please see line 27-28.
Line 66: Ethanol was procured Merck….please revise this (e.g. Ethanol was produced by Merck..)
***The correction has been done to ‘Ethanol was procured from Merck’. Please see line 66.
Line 114: Y is the dependent variables…as Y stands for few variables, should it be Y are dependent variables?
***Yes, there were several dependent variables. Correction has been done. Please see line 115. Thank you for suggestion.
Line 140: please replace this °C with correct one
*** The correction has been done. Please see line 141.
Line 172: abbreviations LC/DAD/MSD and LC/MS have to be explained
*** The abbreviations have been defined. Please see line 172-174.
Lines 174 and 346: numbers of references in brackets and brackets are written in different type of letters
*** The correction has been done throughout the text. Please see line 176 and 364.
Line 192: cm2 should be replaced with cm2
***Corrected. Please see line 194.
Line 198: reference should be converted in the number of the reference and according to that, all the other references need to be checked
****All references have been checked and the numbering has been used for the whole manuscript.
Line 220: This sentence is unclear, please rephrase it. The begging of this section needs to be complemented, meaning there has to be a little entry in the results
*****The first sentence has been provided. Please see line 222-223.
Line 381: make a space between P. aeruginosa
****Corrected. Please see line 406.
Line 470 and 473: there are no dots at the end of those two titles and in the other titles of figures and tables the authors have put the dot at the end
****Dot has been provided in the titles of all Tables and Figures.
References: somewhere the full name of journal is written and somewhere only abbreviation
****All the references have been checked and corrected for consistency. Thank you.
Why there are no dots after the number of the subchapters?
****Sorry for mistake. Dots have been added for all subchapters. Thank you very much.

Reviewer 2 Report
The presented work is devoted to the study of the extraction of phenolic compounds from cashew (Anacardium occidentale L.) leaves and the study of biological activity.
Problem is solved and paper is well presented. The presented work is relevant. Many experiments have been conducted which are mathematically competently described.
Recommendation
1. Specifity the purity of chemicals in section 2.1.
2. Describe the choice of chromatograpgic conditions (why did you choose this mobile phase and column).
3. Conclusion should be rewritten according to obtained results.
4. Add comparative results with reported study.
5. Add results of the study of greeness of analytical method
6. Add system suitability parameters.
Author Response
Responses to reviewer
Comments and Suggestions for Authors
The presented work is devoted to the study of the extraction of phenolic compounds from cashew (Anacardium occidentale L.) leaves and the study of biological activity.
Problem is solved and paper is well presented. The presented work is relevant. Many experiments have been conducted which are mathematically competently described.
**** Thank you so much for the valuable comment and suggestion. All corrections have been made as highlighted in green.
Recommendation
- Specifity the purity of chemicals in section 2.1.
*** Purity of chemicals has been specified. Please see line 66-68.
- Describe the choice of chromatograpgic conditions (why did you choose this mobile phase and column).
***This chromatographic condition to identify and quantify the phenolic compounds was adopted from the same procedure used in previous work.
Reference:
Chotphruethipong, L.; Benjakul, S.; Kijroongrojana, K. Optimization of extraction of antioxidative phenolic compounds from cashew (Anacardium occidentale L.) leaves using response surface methodology. J. Food Biochem. 2017, 41, e12379.
- Conclusion should be rewritten according to obtained results.
****The main/key finding has been already provided in the conclusion in the concise form. Therefore, authors would like to keep the conclusion in the present form.
- Add comparative results with reported study.
*****The comparative results from other studies have been included in the text for most sections. This suggestion is also given by another reviewer. Please see line 313-315, 349-353, 369-372, 392-394, 418-420 and 517-521.
- Add results of the study of greeness of analytical method
******The study on green process, especially decholorphyllization using sedimentation process without the solvent has been provided in the text. Water was added into the extract to remove chlorophylls via precipitation. Please see line 242-246.
- Add system suitability parameters.
*** In the present study, the suitable systematic parameters of phenolic compounds extracted from cashew (Anacardium occidentale L.) leaves has been provided in the conclusion. Please see in line 531-532.
